# Correlation between Electrophysiological Change and Facial Function in Parotid Surgery Patients

**DOI:** 10.3390/jcm10245730

**Published:** 2021-12-07

**Authors:** Feng-Yu Chiang, Chih-Chun Wang, Che-Wei Wu, I-Cheng Lu, Pi-Ying Chang, Yi-Chu Lin, Ching-Feng Lien, Chien-Chung Wang, Tzu-Yen Huang, Tzer-Zen Hwang

**Affiliations:** 1Department of Otolaryngology-Head and Neck Surgery, E-Da Hospital, Kaohsiung 824, Taiwan; fychiang@kmu.edu.tw (F.-Y.C.); ccw5969@yahoo.com.tw (C.-C.W.); lien980206@yahoo.com.tw (C.-F.L.); Kanryw@gmail.com (C.-C.W.); 2School of Medicine, College of Medicine, I-Shou University, Kaohsiung 824, Taiwan; 3Department of Otolaryngology-Head and Neck Surgery, Kaohsiung Medical University Hospital, Faculty of Medicine, College of Medicine, Kaohsiung Medical University, Kaohsiung 807, Taiwan; cwwu@kmu.edu.tw (C.-W.W.); reddust0113@yahoo.com.tw (Y.-C.L.); 4Department of Anesthesiology, Kaohsiung Municipal Siaogang Hospital, Kaohsiung Medical University Hospital, Faculty of Medicine, College of Medicine, Kaohsiung Medical University, Kaohsiung 807, Taiwan; u9251112@gmail.com; 5Department of Anesthesiology, Kaohsiung Municipal Tatung Hospital, Kaohsiung Medical University Hospital, Faculty of Medicine, College of Medicine, Kaohsiung Medical University, Kaohsiung 801, Taiwan; annabelle69@gmail.com

**Keywords:** parotidectomy, facial nerve monitoring (FNM), electromyography (EMG), facial expression, facial grading system

## Abstract

This observational study investigated intraoperative electrophysiological changes and their correlation with postoperative facial expressions in parotidectomy patients with visual confirmation of facial nerve (FN) continuity. Maximal electromyography(EMG) amplitudes of the facial muscles corresponding to temporal, zygomatic, buccal and mandibular branches were compared before/after FN dissection, and facial function at four facial regions were evaluated before/after parotidectomy in 112 patients. Comparisons of 448 pairs of EMG signals revealed at least one signal decrease after FN dissection in 75 (67%) patients. Regional facial weakness was only found in 13 of 16 signals with >50% amplitude decreases. All facial dysfunctions completely recovered within 6 months. EMG amplitude decreases often occur after FN dissection. An amplitude decrease >50% in an FN branch is associated with a high incidence of dysfunction in the corresponding facial region. This study tries to establish a standard facial nerve monitoring (FNM) procedure and a proper facial function grading system for parotid surgery that will be useful for the future study of FNM in parotid surgery.

## 1. Introduction

Facial dysfunction caused by facial nerve (FN) injury is a common and serious complication after surgery for parotid tumors. The reported incidence of facial dysfunction ranges from 14 to 66% for temporary facial weakness and from 0 to 9% for permanent facial weakness [1,2,3,4,5,6,7,8,9,10,11,12,13]. FN injury can cause facial asymmetry, mastication difficulty, drooling, and corneal ulceration. These dysfunctions have severe impacts on the patients’ quality of life and might lead to medical legal litigation.

To preserve FN function, identifying the main trunk of the FN and meticulously dissecting the FN branches are standard procedures in parotid surgery. However, unrecognized nerve injury during FN dissection and unexpected postoperative facial dysfunction or weakness can occur even when anatomical integrity of FN continuity is confirmed intraoperatively.

In recent decades, facial nerve monitoring (FNM) has been popularly applied as an adjunct during parotidectomy for: (i) early identification of the FN trunk; (ii) differentiation of the FN from other tissue and facilitation of FN branch dissection; (iii) confirmation of the functional integrity of the FN; (iv) detection of nerve injury and (v) prognostication of facial expression after resection of the parotid tumor [14,15,16,17,18,19,20,21,22,23,24,25]. However, due to lack of standard procedures for using FNM in parotidectomy, and lack of a proper system for grading facial function after parotidectomy, the value of FNM technology has been debated in the literature. Several studies have reported that use of FNM does not significantly reduce the incidences of temporary and permanent facial dysfunction [14,15,16,17]. Others have reported that FNM decreases the risk of temporary facial weakness but not the risk of permanent facial weakness [18,19,20]. Sajisevi et al. reported that patients who received revised parotidectomy with FNM had shorter operative time, less FN injury severity, and faster FN injury recovery compared with those without FNM [20]. Chiesa-Estomba et al. reported that FNM may decrease the risk of immediate postoperative and permanent FN injury in primary parotidectomy [21]. Ozturk et al. reported that no electrophysiologic parameters were reliable to predict FN dysfunction after superficial partial parotidectomy [22]. Meier et al. also reported that no abnormalities observed by intraoperative continuous FNM during parotidectomy were predictors of FN injury [23]. Nevertheless, Mamelle [24] and Guntinas-Lichius [25] reported that a low ratio of post-dissection to pre-dissection maximal response amplitude was associated with early postoperative facial dysfunction.

In this observational study, we focused on the change of EMG amplitude after FN dissection in patients with intraoperative visual integrity of FN. The EMG amplitudes of four elicited signals represent the function of each FN branch and regional area of facial muscles. When EMG amplitude decreases on one channel, it means that FN branch could be injured. We can map the position of amplitude decrease on the FN branch and detect nerve injury area using a stimulating probe. With this method, we can recognize if nerve injury is on the FN branch or main trunk. The amount of amplitude decrease may predict the regional facial expression after parotidectomy. All surgeries in our study were performed using the same standard procedure for FNM. The electromyography (EMG) amplitudes at four separate regions of facial muscle groups innervated by the temporal, zygomatic, buccal, and marginal mandibular branches of the FN were monitored before and after FN dissection. Additionally, facial function or expression was evaluated by examining individual dynamic movement of muscle groups over four facial regions. This study not only aims to investigate intraoperative changes in EMG amplitude after FN dissection and their correlations with postoperative facial function in patients with visual confirmation of anatomical integrity of FN branches, but also tries to establish a standard FNM procedure and a novel facial grading system for parotid surgery that will be useful for the future study of FNM.

## 2. Materials and Methods

This observational study retrospectively reviewed 120 patients who had undergone primary parotid surgery with FNM from January 2014 to December 2019. Pediatric patients, revision parotidectomy, or patients with bilateral parotid tumors were excluded before patient enrollment. All surgeries were performed by a single surgeon (F-Y, C). Eight parotid cancer patients were excluded due to preoperative facial paralysis or sacrifice of an FN branch during surgery. Thus, this study analyzed 105 benign and seven malignant tumors in 112 patients (51 females and 61 males; mean age, 49.7 ± 13.9 years) with intraoperative visual integrity of FN branches in continuity (Table 1).

All patients were informed of the intent to use FNM to help identify FN trunk and branches and to evaluate FN function during surgery. The authors had no personal, professional, or financial associations with the manufacturer of the nerve-monitoring device used in this study. Ethical approval of this study was obtained from the Kaohsiung Medical University Hospital Institutional Review Board (KMUHIRB-E(I)-20200136). The percentage, mean value, and standard deviations were calculated using Microsoft Excel 2016 (Microsoft Corp., Redmond, WA, USA).

### 2.1. General Anesthesia and Facial Nerve Monitoring Setup

Standard procedures for general anesthesia were performed by the intraoperative neuromonitoring team, which included two experienced anesthesiologists and one nurse anesthetist for anesthetic care and recording. After general anesthesia was induced with lidocaine (1 mg/kg), propofol (2–3 mg/kg); a single dose (0.3 mg/kg) of rocuronium was administered to facilitate endotracheal tube insertion. When maximal neuromuscular blockade was achieved, another bolus of propofol (50 mg) was given. Anesthesia was maintained with sevoflurane and propofol target-controlled infusion. No neuromuscular blocking agents (NMBA) were administered repeatedly during surgery.

After general anesthesia, intraoperative four-channel facial nerve EMG monitoring was performed with a NIM-Response 3.0 system (Medtronic Xomed, Jacksonville, FL, USA) in all patients. Four paired subdermal electrodes (length, 12.0 mm; diameter, 0.4 mm; Medtronic, Jacksonville, FL, USA) were inserted into the lower forehead, infraorbital area, superolateral upper lip and inferolateral lower lip on the ipsilateral side of surgery to monitor activity of facial muscles (Figure 1) innervated by the temporal, zygomatic, buccal, and marginal mandibular branches of the FN. After stimulation of the FN trunk, the elicited EMG signals were displayed on channels 1, 2, 3 and 4 of the monitor screen. The duration of stimulation was 100 µs, the event threshold was 100 µV, event capture was at the largest signal, stimulus current was 3 mA, and frequency was 4 Hz.

### 2.2. Evaluation of FN Function before and after FN Branch Dissection

During surgery, the main trunk of the FN was routinely stimulated with 3 mA when it was first identified. The EMG amplitudes of four elicited signals were used as reference values for FN function and defined as F_1_ signals (Figure 2A,B). The FN branches were then carefully dissected. After dissection of the FN branches and resection of the parotid tumor, the same stimulus current was applied to the FN trunk, and the elicited EMG signals were defined as F_2_ signals (Figure 2C,D). As the EMG amplitudes on each channel of F_2_ and F_1_ signals were compared, they could be classified as unchanged, increased or decreased. An unchanged amplitude was defined as an amplitude with a ±10% change; an increased amplitude was defined as an amplitude increase >10%; a decreased amplitude was defined as an amplitude decrease >10%. All FNs were also photographed after exposure (Table 2).

### 2.3. Assessment of Facial Function

Facial function or expression was assessed by at least two observers before surgery and one day after surgery. Our facial grading system evaluated dynamic movement of individual muscle groups over four separate facial regions (forehead, peri-orbital area, upper mouth, and lower mouth respectively controlled by the temporal, zygomatic, buccal and marginal mandibular branches of the FN). All patients were asked to perform four facial expressions rapidly: wrinkling the forehead, closing the eyes tightly, whistling, and smiling widely. The four facial expressions were photographed before and after surgery (Figure 3). Each of the four facial regions was scored from 3 to 0 points for each facial region according to the different grades of facial expressions. Normal facial function (3 points) was defined as a full symmetric dynamic movement of a facial region. Mild facial dysfunction (2 points) was defined as slightly asymmetrical dynamic movement but symmetrical facial expression. Moderate facial dysfunction (1 point) was defined as obvious asymmetrical dynamic movement and asymmetrical facial expression. Severe facial dysfunction (0 points) was defined as a complete lack of dynamic movement (Table 3). For example, a patient with normal function in each of the four branches would receive a total score 12, T(3)Z(3)B(3)M(3). All the patients with facial dysfunction received postoperative follow-up per month to evaluate the recovery of facial expression.

## 3. Results

The analysis included 448 pairs of EMG signals obtained from the 112 patients in this study. The mean EMG amplitudes of F_2_/F_1_ signals were 983 ± 545/986 ± 592 µV on channel 1, 1289 ± 905/1264 ± 940 µV on channel 2, 1395 ± 961/1462 ± 922 µV on channel 3, and 1542 ± 907/1680 ± 986 µV on channel 4. Comparisons of F_2_ and F_1_ signal amplitudes on each channel showed unchanged amplitude in 223 (50%) signals, increased amplitude in 105 (23%) signals, and decreased amplitude in 120 (27%) signals. Regional facial function was normal in all 328 signals with unchanged or increased amplitudes (Table 1).

The 120 decreased amplitude signals were acquired from 75 patients (67%). Of these 75 patients, 41 patients had 1 decreased signal, 24 patients had 2 decreased signals, 9 patients had 3 decreased signals, and 1 patient had 4 decreased signals. The 120 decreased amplitude signals included 24 in the temporal branch, 19 in the zygomatic branch, 34 in the buccal branch and 43 in the marginal mandibular branch. Amplitude decreases were 10–20% in 42 signals, 20–30% in 32 signals, 30–40% in 23 signals, 40–50% in 7 signals, 50–60% in 2 signals, 60–70% in 7 signals, 70-80% in 4 signals and 80–90% in 3 signals. Regional facial weakness only occurred in 13 of the 16 signals in which amplitude decreases exceeded 50%. Of the thirteen patients with regional facial weakness, seven had mild facial dysfunction, and six had moderate facial dysfunction. The site of regional facial weakness in these thirteen patients was the peri-orbital area in one, the upper mouth in four, and the lower mouth in eight (Table 4). All facial dysfunctions completely recovered within 6 months.

## 4. Discussion

The facial muscles are a group of striated skeletal muscles innervated by the FN that controls facial expression. When FNM is applied in parotid surgery, EMG is used as a diagnostic technique to assess the function of facial muscles and the corresponding FN branches that innervate them. The EMG amplitude is the sum of all differences in the electric potential of all active motor units of a muscle. Measurements of EMG amplitude may correlate with the number of motor units that contribute to polarization [26,27]. Theoretically, an unchanged or increased EMG amplitude after FN branch dissection indicates that postoperative FN function and facial expression will be normal. In this study, all of the 328 signals with unchanged or increased amplitudes showed normal regional facial function after surgery. Conversely, decreased EMG amplitude after FN dissection indicates a nerve function deficit or a decreased number of motor units participating in polarization. However, the correlation between the residual ratio of EMG amplitude after FN injury and the sufficient muscle strength required for normal facial expression is still unknown. In our series, we found the occurrence of EMG amplitude decreases was common after FN branch dissection. In 112 patients with integral continuity of FN branches, 75 patients (67%) had at least one decreased signal after FN dissection. Although 120 signals were detected with decreased amplitude, only 13 of the 16 signals that had an amplitude decrease >50% showed regional facial weakness. These results suggest that patients with a high ratio of amplitude decreases may not have sufficient facial muscle strength for symmetrical facial expression. Mamelle et al. [25] also reported that a low postdissection to predissection ratio of maximal response amplitude was associated with early postoperative facial dysfunction.

The mechanism of FN injury during parotidectomy may include nerve transection, traction, mechanical dissection trauma, ligature entrapment, thermal injury and ischemia. [3] All FN branches in our study were meticulously dissected after identification of the FN trunk. We found mechanical dissection trauma and traction injury were the most important factors for FN injury in patients with ensured anatomical continuity of FN branches. Most mechanical dissection traumas or traction injuries cannot be visually detected. However, in some cases, e.g., parotid cancer with extracapsular extension (Figure 4) or deep lobe tumor (Figure 5), in which FN branches are adhered to the tumor, a nerve branch segment may appear red and swollen after nerve dissection. The severity of nerve injury can be evaluated by comparing EMG amplitudes between F_2_ and F_1_ signals on each channel. A notable finding of this study is that some patients with decreased EMG amplitude after FN dissection had partial or complete intraoperative recovery of EMG. Intraoperative recovery of EMG in these patients was very similar to thyroid surgery patients with traction injury of the recurrent laryngeal nerve (RLN) that showed gradual and progressive recovery of EMG amplitude [28,29,30]. This finding may explain why two patients with substantial amplitude decreases (70% and 85%) still had normal facial expression after surgery. The knowledge of RLN IONM includes the standardization procedure [31], the mechanism of RLN injury [30], and the correlation between electrophysiological change of RLN and vocal cord function [32,33] are helpful for the development of FNM technique. The current study was focused on FNM in parotid surgery. We tried to establish a standard FNM procedure, which may have contribution to the research of FNM such as IONM of RLN in the future, such as (1) to detect where the FN was injured, (2) to elucidate the mechanism of FN injury, (3) to predict the outcome of postoperative facial function, (4) to differentiate a reversible from irreversible FN injury.

The various systems for grading FN function that have been discussed in the literature can be classified as global and regional, or as subjective and objective [34,35,36,37,38,39,40]. The House-Brackmann and Sunnybrook facial grading systems are most used to evaluate facial function. However, the House-Brackmann grading system measures the global function of the facial nerve, which is most applicable in lesions or injuries to the trunk of the facial nerve and not suitable for parotidectomy patients. The Sunnybrook facial grading system evaluates six facial movements (eyebrows, eyelids, nasal base, upper lip, and lower lip). Furthermore, the system globally evaluates resting symmetry, symmetry of voluntary movement, and the degree of synkinesis, which are too complex to execute after parotidectomy. A uniform system for grading facial function after parotidectomy has not been established. Stodulski et al. emphasized that an adequate assessment of FN function after parotidectomy requires a careful assessment of functional deficits in individual FN branches. Ideally, a scale for FN function should be highly reproducible, be quick and easy to perform, and account for the severity of damage to individual FN branches [39]. The current study assessed facial function by examining dynamic movement of individual muscle groups at the four facial regions innervated by temporal, zygomatic, buccal and marginal mandibular branches of the FN. In all patients, four rapid facial expressions were evaluated by at least two observers and photographed before and after surgery. In the 10 patients with facial dysfunction, seven had facial weakness in one region, and three had facial weakness in two regions. The seven patients with slight asymmetry in dynamic movement had mild facial dysfunction. Six patients had moderate facial dysfunction. In all six patients, dysfunction was at the lower mouth area (Table 4). No facial dysfunctions were classified as severe, and all facial dysfunctions completely recovered within 6 months. Based on the results of this study, we conclude that this facial grading system is a simple, quick, reliable and reproducible method of evaluating facial function after parotidectomy.

When FNM is used in parotid surgery, NMBA should be avoided because a neuromuscular blockade will interfere with the interpretation of intraoperative EMG amplitudes [3,19]. However, the absence of adequate neuromuscular blockade during endotracheal intubation increases the risk of laryngeal trauma and the difficulty of intubation [41]. Lu et al. reported that reversal of rocuronium (sugammadex) provides adequate neuromuscular blockade during induction of general anesthesia and does not interfere with interpretation of EMG amplitudes during FNM [42]. However, the high cost of sugammadex limits its routine use in FNM. Lu et al. also reported that the correlated twitch of neuromuscular transmission showed around 73% of recovery at 30 min after administration of 1 ED_95_ of rocuronium (0.3 mg/kg) during intraoperative neuromonitoring of RLN in thyroid surgery, and high EMG signals were obtained in all patients at an early stage of surgery [43]. To ensure safe endotracheal intubation in our study, a low dose (0.3 mg/kg) of rocuronium was administered at induction of general anesthesia, and satisfactory intubation was achieved in most patients. The time from induction of general anesthesia to the first FN trunk stimulation approximated to 60 min. Comparison of F_2_ and F_1_ signals showed that 50% were unchanged, 23% were increased, and 27% were decreased. This result suggested that low-dose rocuronium had a minimal effect on the reference F_1_ signal.

Several limitations of this study are noted:(1)The lack of a control group, good statistical analysis, detailed data regarding type of parotidectomy, tumor size, tumor location and extent of parotid resection and FN dissection were the major shortcoming of this study. However, this study was focused on the change of EMG amplitude and the risk of facial dysfunction after facial nerve dissection with our standard FNM procedure and facial function grading system. Our results showed that regional facial weakness occurred in 13 of 16 signals (81%) with >50% amplitude decrease. An amplitude decrease >50% in an FN branch is associated with a high incidence of dysfunction in the corresponding facial region.(2)Defining an unchanged amplitude after FN dissection is difficult. On each channel, some difference of EMG amplitude was observed between two consecutive stimulations. Therefore, an EMG amplitude change of ±10% may be interpreted as a normal variation of the monitoring system.(3)This study focused on cases with integral continuity of FN branches after parotid tumor resection. Future studies should investigate the outcomes after intraoperative transection of FN branch.(4)The cases with large amplitude decrease were limited, although the results showed a high incidence of facial dysfunction in the signals with amplitude decrease >50%. Further study with large volume in multiple centers is necessary.

## 5. Conclusions

Lack of a standard FNM procedure and uniform facial grading system for parotidectomy has limited the value of FNM in parotid surgery. The standardized FNM procedure and facial grading system used in this observational study are useful not only for identification of the FN trunk and branches, but also for intraoperative assessment of FN function after resection of a tumor, for detection of nerve injury, and for elucidation of an injury mechanism. Therefore, surgeons may use the procedure and grading system applied in this study to assess improvements in surgical technique and to predict facial function outcomes after surgery. Although an EMG amplitude decrease is common after FN dissection in parotid surgery, regional weakness of facial expression rarely occurred in patients with an amplitude decrease < 50% in the corresponding FN branch. However, patients with an amplitude decrease > 50% had a high incidence of facial dysfunction.

## Figures and Tables

**Figure 1 jcm-10-05730-f001:**
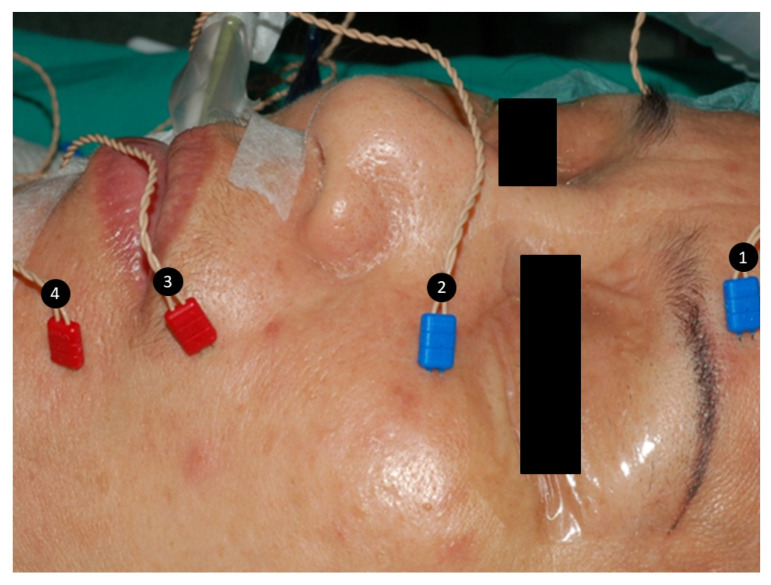
Four paired subdermal electrodes were inserted into the (1) lower forehead, (2) infraorbital area, (3) superolateral upper lip, and (4) inferolateral lower lip to monitor activity of facial muscles innervated by the temporal, zygomatic, buccal, and marginal mandibular branches of the facial nerve, respectively.

**Figure 2 jcm-10-05730-f002:**
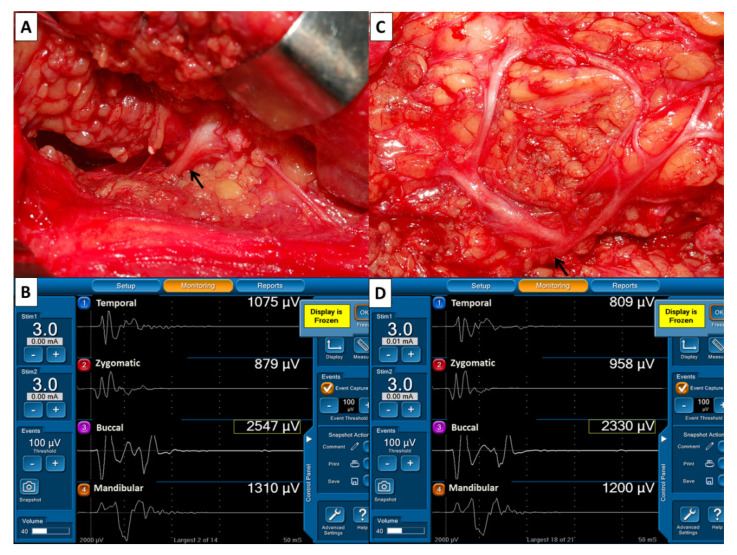
Evaluation of facial nerve (FN) function before and after dissection of FN branches. (**A**) After identification of the main trunk of the FN (↑), a stimulus current of 3 mA was applied. (**B**) The EMG amplitudes of four elicited signals were designated F1 signals and used as reference values for FN function. (**C**) After dissection of FN branches and resection of the parotid tumor, the main trunk of FN (↑) was stimulated with the same stimulus current, and (**D**) the elicited EMG signals were designated F2 signals. On each channel, EMG amplitudes were compared between F2 and F1 signals. This showed 25% of decreased amplitude on channel 1 (809/1075 µV), 11% of increased amplitude on channel 2 (958/879 µV), unchanged amplitude on channel 3 (2330/2547 µV) and channel 4 (1200/1310 µV).

**Figure 3 jcm-10-05730-f003:**
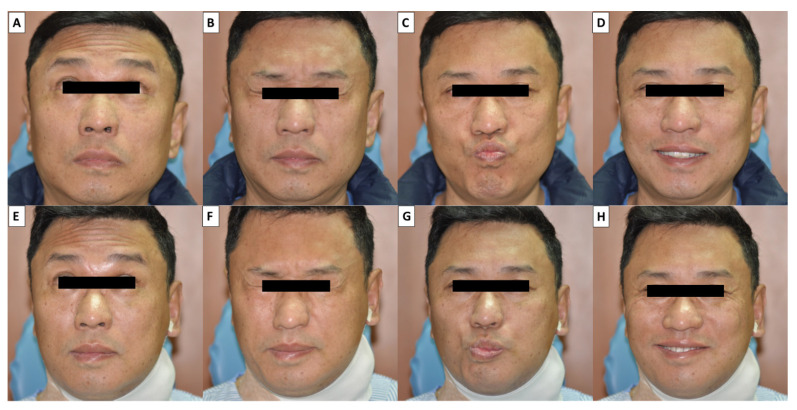
Four facial expressions were photographed before (**A**–**D**) and after operation (**E**–**H**)—wrinkling the forehead (**A**,**E**), closing the eyes tightly (**B**,**F**), whistling (**C**,**G**), and smiling widely (**D**,**H**). Moderate facial dysfunction in the lower part of the mouth was observed during whistling (**G**) and smiling (**H**). (The photos have been approved for publication by the patient).

**Figure 4 jcm-10-05730-f004:**
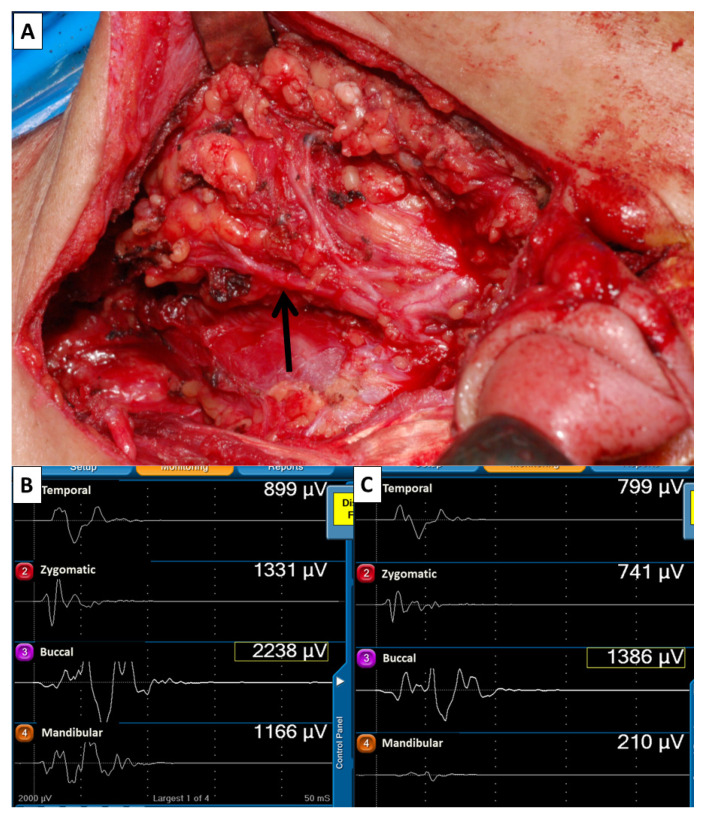
(**A**) Anatomical continuity of FN branches was visually confirmed after meticulous dissection of FN branches in a parotid cancer patient with extracapsular extension (case No.12). A segment of the mandibular branch (↑) had a red swollen appearance. Comparison of EMG amplitudes of F2 (**B**) and F1 (**C**) signals, which showed 11% (799/899 µV), 44% (741/1331 µV), 38% (1386/2238 µV) and 82% (210/1166 µV) of decreased amplitude on channel 1, 2, 3 and 4, respectively. Moderate facial dysfunction was only found in the lower mouth (Figure 3).

**Figure 5 jcm-10-05730-f005:**
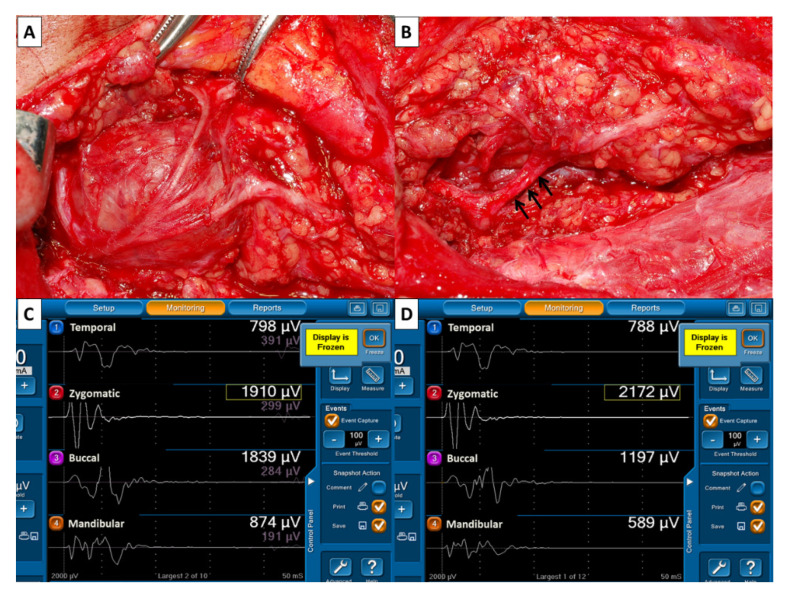
(**A**) The FN branches adhered to the deep lobe parotid tumor. (**B**) A segment of the inferior cervicofacial branch (↑) had a red swollen appearance after tumor resection. Comparison of EMG amplitudes of F2 (**C**) and F1 (**D**) signals revealed unchanged amplitude on channel 1 (788/798 µV) and channel 2 (2172/1970 µV), 35% (1197/1839 µV) and 33% (589/874 µV) of decreased amplitude on channel 3 and 4. The patient had normal facial expression after surgery.

**Table 1 jcm-10-05730-t001:** Characteristics of 112 patients analyzed in current observational study.

Characteristics of Patients	112 Patients (448 NBAR)
Pathologic reports	
Benign	105 patients
Malignant	7 patients
Sex	
Female	51 patients
Male	61 patients
Age (Mean ± SD)	49.7 ± 13.9 years
Surgical extent	
Superficial parotidectomy	89 patients
Superficial and deep parotidectomy	23 patients
Transection facial nerve injury	0 NBAR
Mean EMG amplitude (F2/F1 signals)	
Channel 1	983 ± 545/986 ± 592 µV
Channel 2	1289 ± 905/1264 ± 940 µV
Channel 3	1395 ± 961/1462 ± 922 µV
Channel 4	1542 ± 907/1680 ± 986 µV
Comparisons of F2 and F1 signal amplitudes	
Unchanged	223 (50%) NBAR
Increased	105 (23%) NBAR
Decreased	120 (27%) NBAR
Abnormal regional facial function	
Unchanged/Increased signal amplitude	0 of 328 (0.0%) NBAR
Decreased signal amplitude	
<50% signal amplitude decrease	0 of 104 (0.0%) NBAR
>50% signal amplitude decrease	13 of 16 (81.3%) NBAR

SD = standard deviation; NBAR = Nerve branches at risk; EMG = electromyography.

**Table 2 jcm-10-05730-t002:** Standard procedures of facial nerve monitoring for parotid surgery.

Procedures	Remarks
Grading facial function and photographing the four facial expressions before surgery	Evaluate dynamic movement of individual muscle groups over four separate facial regions by performing four rapid facial expressions
General anesthesia and facial nerve monitoring setup	Only a single dose (0.3 mg/kg) of rocuronium is administered during induction of general anesthesia. Four paired subdermal electrodes are inserted into four separate regions to monitor activity of facial muscles innervated by the temporal, zygomatic, buccal, and marginal mandibular branches of FN.
Pre-dissection EMG (F1 signals)	FN trunk is stimulated with 3 mA when it is first identified
Post-dissection EMG (F2 signals)	After dissection of the FN branches, 3 mA is applied to FN trunk
Interpretation of EMG signals (F2/F1 ratio)	Unchanged amplitude- amplitude change within ±10% Increased amplitude- amplitude increase >10% Decreased amplitude- amplitude decrease >10%
Photographing the exposed FN branches	Confirmation of visual integrity of the FN
Grading facial function and photographing the four facial expressions after surgery	If asymmetric facial expression is detected, compare with preoperative recording

**Table 3 jcm-10-05730-t003:** Facial grading system evaluating dynamic movement of facial expressions over four separate facial regions.

Degree	Description	Points
Normal function	A full symmetric dynamic movement of a facial region	3
Mild dysfunction	A slightly asymmetrical dynamic movement but symmetrical facial expression	2
Moderate dysfunction	An obvious asymmetrical dynamic movement and asymmetrical facial expression	1
Severe dysfunction	A complete lack of dynamic movement	0

**Table 4 jcm-10-05730-t004:** Assessment of regional facial function in 13 patients with EMG amplitude decrease > 50%.

Case Number	Age, Sex Side	Branches of Facial Nerve	Amplitude Decrease (%)	Grade of Facial Dysfunction	Pathologic Report
Case 1	74, M Left	Zygomatic	85%	Normal	Warthin’s tumor
Case 2	42, F Left	Zygomatic Buccal	75% 53%	Mild Mild	Pleomorphic adenoma
Case 3	58, M Right	Buccal	66%	Mild	Pleomorphic adenoma
Case 4	47, M Right	Temporal	53%	Normal	Pleomorphic adenoma
Case 5	47, F Left	Buccal, Mandibular	75% 61%	Mild Moderate	Warthin’s tumor
Case 6	70, F Right	Mandibular	69%	Mild	Lymphoepithelial cyst
Case 7	67, M Left	Mandibular	68%	Moderate	Warthin’s tumor
Case 8	55, M Right	Zygomatic	70%	Normal	Hemangioma
Case 9	39, F Right	Buccal Mandibular	62% 73%	Mild Moderate	Myoepithelioma
Case 10	53, M Left	Mandibular	63%	Mild	Warthin’s tumor
Case 11	38, F Right	Mandibular	61%	Moderate	Oncocytoma
Case 12	48, M Left	Mandibular	82%	Moderate	Salivary duct carcinoma
Case 13	54, M Right	Mandibular	90%	Moderate	Warthin’s tumor

## Data Availability

The original contributions presented in the study are included in the article. Further inquiries can be directed to the corresponding authors.

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
