# Peer review of "Correlation between Electrophysiological Change and Facial Function in Parotid Surgery Patients"

_jcm, 2021, doi:10.3390/jcm10245730_

Round 1

Reviewer 1 Report

Dear Authors

You have nicely presented a very interesting topic that has been covered over the last 2 decades. My only - severe- reservation is that you dont add any information about the type of surgery for any of your patients, although the study covers a relatively recent period (2014-2019). We can not deduce whether for example an area with high amount of interconnecting branches (buccal) had significant amount of weakness. Also, is weakness stemming from injury to main trunk or its arborisations.

Another objection is why you chose a new facial grading system instead of well established ones (House-Brackmann, Sunnybrook). 

Author Response

Author's Response

Dear Reviewer,

We deeply appreciate your comments.

We have revised our manuscript in-line with the comment made.

The followings are our response:

Response to the Reviewer #1

Comment-1:

You have nicely presented a very interesting topic that has been covered over the last 2 decades.

Response:

Thank you for the comment. We totally agree that the topic of facial nerve monitoring (FNM) in parotid surgery had been covered over the recent decades in the literature. However, due to lack of standard procedures for using FNM in parotidectomy and proper system for grading facial function after parotidectomy, the value of FNM technology is limited.

This study tries to establish a standard FNM procedure and a novel facial function grading system for parotid surgery that will be useful for the future study of FNM in parotid surgery.

Comment-1:

    My only - severe- reservation is that you dont add any information about the type of surgery for any of your patients, although the study covers a relatively recent period (2014-2019). We can not deduce whether for example an area with high amount of interconnecting branches (buccal) had significant amount of weakness. Also, is weakness stemming from injury to main trunk or its arborisations.

Response:

        Thank you for the comment. We focused this study on the change of EMG amplitude after FN dissection in the patients with intraoperative visual integrity of FN. The EMG amplitudes of four elicited signals represent the function of each FN branch and regional area of facial muscles. When EMG amplitude decreases on one channel, it means that FN branch could be injured. We can map the position of amplitude decrease on the FN branch and detect nerve injury area by stimulating probe. With this method, we can recognize the nerve injury is on the FN branch or main trunk. The amount of amplitude decrease may predict the regional facial expression after parotidectomy.

        We added the description in Discussion/Limitation section. Thank you for this helpful suggestion.

Comment-2:

    Another objection is why you chose a new facial grading system instead of well established ones (House-Brackmann, Sunnybrook).

Response:

Thank you for the comment. We agree that the House-Brackmann and Sunnybrook facial grading system are most used to evaluate the facial function.

However, the House-Brackmann grading system measures the global function of the facial nerve, which is most applicable in lesions or injuries to the trunk of the facial nerve and not suitable for parotidectomy patients.

The Sunnybrook facial grading system evaluates six facial movements (eyebrows, eyelids, nasal base, upper lip, and lower lip). Furthermore, the system globally evaluates resting symmetry, symmetry of voluntary movement, and the degree of synkinesis, which are too complex to execute after parotidectomy.

The current study assessed facial function by examining dynamic movement of individual muscle groups at the four facial regions innervated by temporal, zygomatic, buccal and marginal mandibular branches of the FN. In all patients, the four rapid facial expressions were evaluated by at least two observers and photographed before and after surgery. The facial grading system used in this study to evaluate the severity of damage to individual FN branches is a simple, quick, reliable and reproductive method to perform.

The reasons why we do not use the House-Brackmann and Sunnybrook facial grading system were added in the Discussion section.

We thank you for your valued comments and suggestions, which we feel substantially improve our manuscript, and hope that the revisions meet with your approval.

Sincerely,

Feng-Yu Chiang, M.D., Tzer-Zen Hwang, M.D.

Department of Otolaryngology-Head and Neck Surgery, E-Da Hospital, I-Shou University, Kaohsiung, Taiwan

Address: No. 1, Yida Road, Jiaosu Village, Yanchao District, Kaohsiung 824, Taiwan.

E-mails: fychiang@kmu.edu.tw

Tzu-Yen Huang, M.D.

Department of Otorhinolaryngology–Head and Neck Surgery, Kaohsiung Medical University Hospital, Kaohsiung Medical University, Kaohsiung, Taiwan.

Address: 100TzYou 1st Road, Kaohsiung 807, Taiwan.

E-mails: tyhuang.ent@gmail.com

 (On behalf of all coauthors)

Nov 7, 2021

Reviewer 2 Report

Thank you for submitting your manuscript to JCM. The authors conducted a retrospective study investigated intraoperative electrophysiological changes and their correlation with postoperative facial expressions in parotidectomy patients with visual confirmation of facial nerve (FN) continuity.

- The study is very limited in the interpretation. The lack of both controlled group and powerful statistical analysis are the major limitations of this study.  What is the novelty or strength of the current study? This should be clearly mentioned in the introduction section.

-Several cofounders effect on the outcomes of interest were not evaluated. Such as, type of parotidectomy,  https://pubmed.ncbi.nlm.nih.gov/34288212/, tumor size and entity, …. etc.  I suggest to do an independent analysis based on the extent of parotid resection (ECD, partial superficial, superficial or total parotidectomy)  the method of facial nerve dissection (Antegrade versus retrograde), tumor entity (malignant versus benign), tumor size, postoperative radiotherapy …etc.

-  Please remove these sentences from conclusion “To avoid permanent facial dysfunction, preserving anatomical continuity of FN branches is essential” and "No patients with visual anatomical integrity of FN had permanent facial dysfunction, which indicates that meticulous dissection of FN branches and preservation of their anatomical continuity are essential in parotid surgery". They are redundant and should be deleted from conclusion.

- What was the exclusion criteria? e.g. Pediatric patient, revision parotidectomy, and patients with bilateral parotid tumors.

-The authors stated that, “All facial dysfunctions completely recovered within 6 months”, What was the mean and range of postoperative Follow-up?

-How facial nerve function was graded?  House-Brackmann classification or Sunnybrook scales.

-How the statistical analysis was conducted?

Author Response

Author's Response

Dear Reviewer,

We deeply appreciate your comments.

We have revised our manuscript in-line with the comment made.

The followings are our response:

Response to the Reviewer #2

Thank you for submitting your manuscript to JCM. The authors conducted a retrospective study investigated intraoperative electrophysiological changes and their correlation with postoperative facial expressions in parotidectomy patients with visual confirmation of facial nerve (FN) continuity.

Comment-1:

    -The study is very limited in the interpretation. The lack of both controlled group and powerful statistical analysis are the major limitations of this study. 

    -How the statistical analysis was conducted?

Response:

Thank you for the precious comment. We agree that the lack of both controlled group and powerful statistical analysis are the major limitations of this study.  However, this study was focused on the change of EMG amplitude and the risk of facial dysfunction after facial nerve dissection with our standard FNM procedure and facial function grading system. Our results showed that regional facial weakness occurred in 13 of 16 signals (81%) with >50% amplitude decrease. An amplitude decrease >50% in an FN branch is associated with a high incidence of dysfunction in the corresponding facial region.

We added the description in Discussion/Limitation section. Thank you for this helpful suggestion.

Comment-2:

    What is the novelty or strength of the current study? This should be clearly mentioned in the introduction section.

Response:

        Thank you for your suggestion, we totally agree that the novelty and strength of the current study should be more clearly mentioned in the manuscript.

The FNM has been used in parotid surgery for several decades, but the value of FNM technology has still been debated in the literature due to lack of standard procedures for using FNM in parotidectomy and lack of proper system for grading facial function after parotidectomy. Therefore, a standard FNM procedure and proper facial grading system for parotid surgery are important issues in this study.

In Abstract, we added the description “This study tries to establish a standard facial nerve monitoring (FNM) procedure and a proper facial function grading system for parotid surgery that will be useful for the future study of FNM in parotid surgery.

We strengthen the novelty of the current study in the Introduction section as following: “In this study, we focused on the change of EMG amplitude after FN dissection in the patients with intraoperative visual integrity of FN. The EMG amplitudes of four elicited signals represent the function of each FN branch and regional area of facial muscles. When EMG amplitude decreases on one channel, it means that FN branch could be injured. We can map the position of amplitude decrease on the FN branch and detect nerve injury area by stimulating probe. With this method, we can recognize the nerve injury is on the FN branch or main trunk. The amount of amplitude decrease may predict the regional facial expression after parotidectomy.“ and “This study not only aims to investigate…, but also tries to establish a standard FNM procedure and a novel facial grading system for parotid surgery that will be useful for the future study of FNM.

Comment-3:

-Several cofounders effect on the outcomes of interest were not evaluated. Such as, type of parotidectomy,  https://pubmed.ncbi.nlm.nih.gov/34288212/, tumor size and entity, …. etc.  I suggest to do an independent analysis based on the extent of parotid resection (ECD, partial superficial, superficial or total parotidectomy)  the method of facial nerve dissection (Antegrade versus retrograde), tumor entity (malignant versus benign), tumor size, postoperative radiotherapy …etc.

Response:

        Thank you for the comment. We totally agree with you that this study has several limitations, including lack of the type of surgery, tumor size, extent of parotid resection and facial nerve dissection, which have been mentioned in the text.

However, this study was focused on FNM in parotid surgery. We try to establish a standard FNM procedure, which may have contribution to the research of FNM like IONM of RLN in the future, such as 1) to detect where the FN was injured, 2) to elucidate the mechanism of FN injury, 3) to predict the outcome of postoperative facial function, 4) to differentiate a reversible from irreversible FN injury, etc.

We start applying nerve monitoring in our thyroid and parotid surgeries since 2006. We have a lot of studies and publications on intraoperative neuromonitoring (IONM) of recurrent laryngeal nerve (RLN) in thyroid surgery, including standardization of IONM of RLN (1), the mechanism of RLN injury with the application of IONM (2), correlation between electrophysiological change of RLN and vocal cord function (3,4), etc.

(1)       Standardization of intraoperative neuromonitoring of recurrent laryngeal nerve in thyroid operation. Chiang FY, Lee KW, Chen HC, Chen HY, Lu IC, Kuo WR, Hsieh MC, Wu CW. World J Surg. 2010 Feb;34(2):223-9.

(2)       The mechanism of recurrent laryngeal nerve injury during thyroid surgery--the application of intraoperative neuromonitoring. Chiang FY, Lu IC, Kuo WR, Lee KW, Chang NC, Wu CW: Surgery. 2008 Jun;143(6):743-9.

(3)       Correlation Between Electrophysiological Changes and Outcomes of Vocal Cord Function in 1764 Recurrent Laryngeal Nerves with Visual Integrity During Thyroidectomy. Yuan Q, Wu G, Hou J, Liao X, Liao Y, Chiang FY. Thyroid. 2020 May;30(5):739-745.

(4)       Recurrent laryngeal nerve injury with incomplete loss of electromyography signal during monitored thyroidectomy-evaluation and outcome. Wu CW, Hao M, Tian M, Dionigi G, Tufano RP, Kim HY, Jung KY, Liu X, Sun H, Lu IC, Chang PY, Chiang FY. Langenbecks Arch Surg. 2017 Jun;402(4):691-699.

        We added the description in Discussion section. Thank you for mentioning this important issue.

Comment-4:

- What was the exclusion criteria? e.g. Pediatric patient, revision parotidectomy, and patients with bilateral parotid tumors.

Response:

        Thank you for your comment and reminder. We added the description about exclusion criteria in Materials and Methods section as “pediatric patients, revision parotidectomy, or patients with bilateral parotid tumors was excluded before patient’s enrollment.

Comment-5:

-  Please remove these sentences from conclusion “To avoid permanent facial dysfunction, preserving anatomical continuity of FN branches is essential” and "No patients with visual anatomical integrity of FN had permanent facial dysfunction, which indicates that meticulous dissection of FN branches and preservation of their anatomical continuity are essential in parotid surgery". They are redundant and should be deleted from conclusion.

Response:

Thank you for the comment. We agree that the paragraph is redundant and have been deleted.

Comment-6:

-The authors stated that, “All facial dysfunctions completely recovered within 6 months”, What was the mean and range of postoperative Follow-up?

Response:

Thank you for your useful comment. We added the description in Materials/Methods section, 2.3 Assessment of facial function as “All the patients with facial dysfunction received postoperative follow-up per month to evaluate the recovery of facial expression.” Thank your again for your previous suggestion.

Comment-7:

-How facial nerve function was graded?  House-Brackmann classification or Sunnybrook scales.

Response:

Thank you for the comment. We agree that the House-Brackmann and Sunnybrook facial grading system are most used to evaluate the facial function.

However, the House-Brackmann grading system measures the global function of the facial nerve, which is most applicable in lesions or injuries to the trunk of the facial nerve and not suitable for parotidectomy patients.

The Sunnybrook facial grading system evaluates six facial movements (eyebrows, eyelids, nasal base, upper lip, and lower lip). Furthermore, the system globally evaluates resting symmetry, symmetry of voluntary movement, and the degree of synkinesis, which are too complex to execute after parotidectomy.

The current study assessed facial function by examining dynamic movement of individual muscle groups at the four facial regions innervated by temporal, zygomatic, buccal and marginal mandibular branches of the FN. In all patients, the four rapid facial expressions were evaluated by at least two observers and photographed before and after surgery. The facial grading system used in this study to evaluate the severity of damage to individual FN branches is a simple, quick, reliable and reproductive method to perform.

The reasons why we do not use the House-Brackmann and Sunnybrook facial grading system were added in the Discussion section.

We thank you for your valued comments and suggestions, which we feel substantially improve our manuscript, and hope that the revisions meet with your approval.

Sincerely,

Feng-Yu Chiang, M.D., Tzer-Zen Hwang, M.D.

Department of Otolaryngology-Head and Neck Surgery, E-Da Hospital, I-Shou University, Kaohsiung, Taiwan

Address: No. 1, Yida Road, Jiaosu Village, Yanchao District, Kaohsiung 824, Taiwan.

E-mails: fychiang@kmu.edu.tw

Tzu-Yen Huang, M.D.

Department of Otorhinolaryngology–Head and Neck Surgery, Kaohsiung Medical University Hospital, Kaohsiung Medical University, Kaohsiung, Taiwan.

Address: 100TzYou 1st Road, Kaohsiung 807, Taiwan.

E-mails: tyhuang.ent@gmail.com

 (On behalf of all coauthors)

Nov 7, 2021

Reviewer 3 Report

This manuscript is a retrospective series on the correlation between electrophysiological change and facial function in parotid surgery patients.

It is an interesting paper, well-written and with excellent figures and results.

I have only two suggestions to improve it:

1.- The authors have mentioned the systematic review on facial nerve monitoring and parotid surgery by Sood et al. (ref. 19).

They should also mention recent ones:

Sajisevi M. Indications for Facial Nerve Monitoring During Parotidectomy. Otolaryngol Clin North Am. 2021 Jun;54(3):489-496. doi: 10.1016/j.otc.2021.02.001.

Chiesa-Estomba CM, Larruscain-Sarasola E, Lechien JR, Mouawad F, Calvo-Henriquez C, Diom ES, Ramirez A, Ayad T. Facial nerve monitoring during parotid gland surgery: a systematic review and meta-analysis. Eur Arch Otorhinolaryngol. 2021 Apr;278(4):933-943. doi: 10.1007/s00405-020-06188-0. Epub 2020 Jul 11.

2.- As the authors comment on lines 313-15, “Detailed data regarding type of parotidectomy, tumor size, tumor location and extent 313 of parotid resection and FN dissection were not collected.”

They should state how many superficial parotidectomies were done exposing the four main branches of the nerve.

Author Response

Author's Response

Dear Reviewer,

We deeply appreciate your comments.

We have revised our manuscript in-line with the comment made.

The followings are our response:

Response to the Reviewer #3

This manuscript is a retrospective series on the correlation between electrophysiological change and facial function in parotid surgery patients.

It is an interesting paper, well-written and with excellent figures and results.

I have only two suggestions to improve it:

Comment-1:

1.- The authors have mentioned the systematic review on facial nerve monitoring and parotid surgery by Sood et al. (ref. 19).

They should also mention recent ones:

Sajisevi M. Indications for Facial Nerve Monitoring During Parotidectomy. Otolaryngol Clin North Am. 2021 Jun;54(3):489-496. doi: 10.1016/j.otc.2021.02.001.

Chiesa-Estomba CM, Larruscain-Sarasola E, Lechien JR, Mouawad F, Calvo-Henriquez C, Diom ES, Ramirez A, Ayad T. Facial nerve monitoring during parotid gland surgery: a systematic review and meta-analysis. Eur Arch Otorhinolaryngol. 2021 Apr;278(4):933-943. doi: 10.1007/s00405-020-06188-0. Epub 2020 Jul 11.

Response:

        Thank you for your precious comment. We added the description in Introduction section as, “ Sajisevi et al. reported that patients received revised parotidectomy with FNM had shorter operative time, less FN injury severity, and faster FN injury recovery compared with those without FNM [20]. Chiesa-Estomba et al. reported that FNM may decrease the risk of immediate postoperative and permanent FN injury in primary parotidectomy [21] “. Both the reference you suggested were mentioned in the revision. Thank you again for your helpful suggestion.

Comment-2:

2.- As the authors comment on lines 313-15, “Detailed data regarding type of parotidectomy, tumor size, tumor location and extent 313 of parotid resection and FN dissection were not collected.”

They should state how many superficial parotidectomies were done exposing the four main branches of the nerve.

Response:

Thank you for your suggestion. In this study we focused on the change of EMG amplitude after FN dissection in the patients with intraoperative visual integrity of FN. When EMG amplitude decreases on one channel, it means that FN branch could be injured. We can map the position of amplitude decrease on the FN branch and detect nerve injury area by stimulating probe. With this method, we can recognize the nerve injury is on the FN branch or main trunk. The amount of amplitude decrease may predict the regional facial expression after parotidectomy.

We understand that this observational study had some major shortcoming, and we modified the description in Discussion/Limitation section as “1) The lack of both control group, good statistical analysis, detailed data regarding type of parotidectomy, tumor size, tumor location and extent of parotid resection and FN dissection were the major shortcoming of this study.”  

In Table 1, we also added the data about surgical extent. Thank you again for your precious comment.

We thank you for your valued comments and suggestions, which we feel substantially improve our manuscript, and hope that the revisions meet with your approval.

Sincerely,

Feng-Yu Chiang, M.D., Tzer-Zen Hwang, M.D.

Department of Otolaryngology-Head and Neck Surgery, E-Da Hospital, I-Shou University, Kaohsiung, Taiwan

Address: No. 1, Yida Road, Jiaosu Village, Yanchao District, Kaohsiung 824, Taiwan.

E-mails: fychiang@kmu.edu.tw

Tzu-Yen Huang, M.D.

Department of Otorhinolaryngology–Head and Neck Surgery, Kaohsiung Medical University Hospital, Kaohsiung Medical University, Kaohsiung, Taiwan.

Address: 100TzYou 1st Road, Kaohsiung 807, Taiwan.

E-mails: tyhuang.ent@gmail.com

 (On behalf of all coauthors)

Nov 30, 2021

Round 2

Reviewer 1 Report

This study is observational and the authors should acknowledge this. Statistical analysis should be performed even on these grounds. Data for patients treated should be added. 

Author Response

Author's Response

Dear Reviewer,

We deeply appreciate your comments.

We have revised our manuscript in-line with the comment made.

The followings are our response:

Response (Round 2) to the Reviewer #1

Comment-1:

This study is observational and the authors should acknowledge this. Statistical analysis should be performed even on these grounds. Data for patients treated should be added.

Response:

Thank you for your comment. We added “observational” in Abstract, Introduction, Materials and Methods, and Conclusions as your suggestion.

        For better demonstration the characteristics of patients, we added the Table 1 to summarize it. The statistical description was also added in Methods and Materials section (Line 107-109). Thank you again for your precious suggestion.

We thank you for your valued comments and suggestions, which we feel substantially improve our manuscript, and hope that the revisions meet with your approval.

Sincerely,

Feng-Yu Chiang, M.D., Tzer-Zen Hwang, M.D.

Department of Otolaryngology-Head and Neck Surgery, E-Da Hospital, I-Shou University, Kaohsiung, Taiwan

Address: No. 1, Yida Road, Jiaosu Village, Yanchao District, Kaohsiung 824, Taiwan.

E-mails: fychiang@kmu.edu.tw

Tzu-Yen Huang, M.D.

Department of Otorhinolaryngology–Head and Neck Surgery, Kaohsiung Medical University Hospital, Kaohsiung Medical University, Kaohsiung, Taiwan.

Address: 100TzYou 1st Road, Kaohsiung 807, Taiwan.

E-mails: tyhuang.ent@gmail.com

 (On behalf of all coauthors)

Nov 30, 2021

Reviewer 2 Report

The patients included in the current study were heterogenous as there were different types of parotidecomies, different tumor size, site, and different tumor pathology. Therefore authors should provide additional analysis evaluation the above mentioned cofounders on the outcome of intrest. Also, the lack of both control group and good statistical analysis were the major shortcoming of this study.

Author Response

Author's Response

Dear Reviewer,

We deeply appreciate your comments.

We have revised our manuscript in-line with the comment made.

The followings are our response:

Response (Round 2) to the Reviewer #2

Comment-1:

The patients included in the current study were heterogenous as there were different types of parotidecomies, different tumor size, site, and different tumor pathology. Therefore authors should provide additional analysis evaluation the above mentioned cofounders on the outcome of intrest. Also, the lack of both control group and good statistical analysis were the major shortcoming of this study.

Response:

Thank you for your suggestion. In this study we focused on the change of EMG amplitude after FN dissection in the patients with intraoperative visual integrity of FN. When EMG amplitude decreases on one channel, it means that FN branch could be injured. We can map the position of amplitude decrease on the FN branch and detect nerve injury area by stimulating probe. With this method, we can recognize the nerve injury is on the FN branch or main trunk. The amount of amplitude decrease may predict the regional facial expression after parotidectomy.

We understand that this observational study had some major shortcoming, and we modified the description in Discussion/Limitation section as “1)      The lack of both control group, good statistical analysis, detailed data regarding type of parotidectomy, tumor size, tumor location and extent of parotid resection and FN dissection were the major shortcoming of this study.

        For better demonstration the characteristics of patients, we added the Table 1 to summarize it. In Table 1, we also added the data of surgical extent. Thank you again for your precious comment.

We thank you for your valued comments and suggestions, which we feel substantially improve our manuscript, and hope that the revisions meet with your approval.

Sincerely,

Feng-Yu Chiang, M.D., Tzer-Zen Hwang, M.D.

Department of Otolaryngology-Head and Neck Surgery, E-Da Hospital, I-Shou University, Kaohsiung, Taiwan

Address: No. 1, Yida Road, Jiaosu Village, Yanchao District, Kaohsiung 824, Taiwan.

E-mails: fychiang@kmu.edu.tw

Tzu-Yen Huang, M.D.

Department of Otorhinolaryngology–Head and Neck Surgery, Kaohsiung Medical University Hospital, Kaohsiung Medical University, Kaohsiung, Taiwan.

Address: 100TzYou 1st Road, Kaohsiung 807, Taiwan.

E-mails: tyhuang.ent@gmail.com

 (On behalf of all coauthors)

Nov 30, 2021